# High Interfacial Adsorption of Light Gases on Nano-Thin Molten Polyethylene Films

**DOI:** 10.3390/polym17202751

**Published:** 2025-10-14

**Authors:** Roberto Guerra-González, Martha A. Lemus-Solorio, Alfonso Lemus-Solorio, José L. Rivera

**Affiliations:** 1Department of Chemical Engineering, Universidad Michoacana de San Nicolás de Hidalgo, Morelia 58000, Mexico; roberto.guerra@umich.mx (R.G.-G.); 1209689x@umich.mx (A.L.-S.); 2Department of Physico-Mathematical Sciences, Universidad Michoacana de San Nicolás de Hidalgo, Morelia 58000, Mexico; martha.lemus@umich.mx

**Keywords:** thin films, polyethylene, ethane, interfacial adsorption, interfacial tension

## Abstract

Classical Molecular Dynamics simulations were used to investigate the interfacial adsorption of supercritical ethane on ultrathin molten polyethylene films at various temperatures (298.15–448.15 K) and pressures (0.28–13.17 MPa). Ethane was found to accumulate preferentially at the film’s interfaces rather than dissolving into the film’s core. The ultra-thin, metastable films, studied at their mechanical stability limit, are composed of two overlapping interfaces. The films show some fractions of interfacial chains transiently desorbing from the film surface and entering the gas phase, which facilitates the accumulation of ethane at the interfaces. At 373.15 K and pressures between 0.29 MPa and 9.65 MPa, the combined film interfaces adsorb between 4.8 and 8.6 times more ethane than the amount solubilized in the central, bulk region of the film. Interfacial tension of the film decreases exponentially with increasing gas pressure of ethane and is primarily governed by inter-chain interactions at the interface. Minor contributions arise from the vibrational dynamics of polyethylene chain fractions that transiently desorb from the film surface. Furthermore, the solubility of ethane in the film’s bulk region exhibits a temperature-dependent inversion: at 298.15 K, the ethane density in the film’s center slightly exceeds that of the bulk gas, but this trend reverses at 373.15 K and becomes more pronounced as the temperature increases. This indicates a potential solubility transition temperature between 298.15 K and 373.15 K.

## 1. Introduction

Improved adsorption materials for ethane, such as engineered Metal–Organic Frameworks [1,2,3] and specialized zeolites [4,5,6], primarily target energy-efficient industrial gas separation in two major areas. First, they aim to replace highly energy-intensive cryogenic distillation in the petrochemical industry by developing ethane-selective adsorbents to purify ethylene, the key building block for plastics, from its ethane precursor [2,4,5]. By preferentially trapping the ethane impurity, the process becomes significantly simpler and more cost-effective. Second, these materials are vital for natural gas processing and natural gas–liquid recovery, enabling the selective removal and capture of valuable ethane from the primary methane stream [1,3,6]. Ethane is also a significant atmospheric pollutant, primarily classified as a volatile organic compound, not for direct toxicity, but due to its role as a precursor to ground-level ozone, a major component of smog that harms human respiratory health and vegetation [7,8]. Composite materials made of zeolite and activated carbon, have shown an enhanced mass transfer efficiency for ethane adsorption from air [9].

Polymeric structures are also employed as adsorbent materials. Specifically, bulk polyethylene (PE) in its linear (HDPE and LLDPE) and branched (LDPE) structures, has been extensively studied for its ability to store and separate various gases of industrial interest, such as H_2_, CO_2_, ethylene, and light alkanes [10,11,12], as well as environmental pollutants like CO_2_, polycyclic aromatic hydrocarbons, and polychlorinated biphenyls [13,14,15]. PE is widely used due to a combination of low cost, excellent processability, high chemical inertness, remarkable durability and flexibility, low moisture absorption, biocompatibility, and electrical insulation properties [16,17,18].

Experimental studies, corroborated by the Statistical Associating Fluid Theory (SAFT), have investigated the capacity of bulk LLDPE to adsorb and dissolve light gases such as ethylene, propene, but-1-ene, and propane [19,20]. At temperatures where LLDPE is in a molten state (413.15−473.15 K), the dissolution capacity for these gases is considerable and increases with the gas’s molecular weight. At 413.5 K, the solubility slightly exceeds 200 g per 1000 g of LLDPE at gas phase pressures of approximately 25.0 MPa (ethene), 5.3 MPa (propene), and 2.5 MPa (but-1-ene). For propane, the only alkane studied, achieving the same solubility required a slightly higher pressure (~ 6.5 MPa), which suggests that smaller alkenes like ethylene may be more soluble than its corresponding alkane (ethane). This trend of solubility as a function of gas molecular weight has also been reported in experimental studies comparing the solubilities of *n*-pentane at 423.65 K [21] and propane at 443.15 K [19], which has been corroborated by close SAFT predictions [22]. Studies on the solubility of ethylene in different bulk PE structures at temperatures between 333.15 K and 423.15 K have found that the degree of crystallinity plays a significant role: higher crystallinity leads to lower solubility [23].

Experimental investigations of PE films have shown that the solubility of pure ethylene increases exponentially with gas pressure, while the solubility of methane and nitrogen increases asymptotically. Methane/ethylene mixtures at 50% by weight behave similarly to pure ethylene [12]. The maximum observed solubility was low, at approximately 67 g of ethylene per 1000 g of PE, at 9 MPa and 298.15 K. The PE films used in these studies were pre-treated with *n*-heptane, and their thickness was not determined, potentially influencing the reported solubilities [24]. The adsorption of CO_2_ and H_2_ has also been studied experimentally in HDPE films without determining their thickness, finding high CO_2_ adsorption (624.3 g/1000 g of HDPE) at 0.3 MPa and 303.15 K, while H_2_ reached only 36.1 g/1000 g of HDPE [10].

In previous work, classical Molecular Dynamics (MD) simulations were used to characterize linear PE films of 200 monomers at their stability limit [25,26]. This allowed us to determine the critical thickness at which a PE film becomes unstable by developing pores. Above this critical thickness, instability pores also develop and typically self-heal within picoseconds. Below it, they grow, causing the film to fragment into aerosol droplets [27,28]. The limiting thicknesses obtained with MD are a few nanometers, slightly lower than those obtained in subsequent state-of-the-art experimental studies where PE films are produced through careful stretching processes to prevent rupture [29]. The pores that destabilize the PE film become more pronounced as the thickness approaches the critical stability value. These pores can be filled with smaller species that are affine to PE, with the goal of increasing stability and potentially further reducing the critical thickness. Alternatively, they can be used as storage sites for problematic light gases or for those of specific commercial interest.

This work focuses on the study of the storage capacity of supercritical ethane adsorbed in very thin, self-standing molten PE films at thicknesses at their stability limit. The following sections detail the classical MD methodology used and present the results and their discussion. This study also includes analyses of the total storage capacity, estimates of ethane solubility in the PE film, the large adsorption of ethane at the interfaces, the volumetric properties of the ethane gas, the interfacial properties of the film with dissolved and adsorbed ethane gas, and their dependence on gas pressure and system temperature. The article concludes with a summary of the main findings and future work.

## 2. Methodology

Classical MD simulations were used to study the interfacial adsorption and solubility of supercritical ethane in thin films formed by molten PE chains. The study focused on films at their mechanical stability limit and at temperatures of 298.15 K, 373.15 K, and 448.15 K. This method is commonly used to simulate various properties of polymeric systems [30,31,32]. The MD code implemented in the Large-scale Atomic/Molecular Massively Parallel Simulator (LAMMPS) [33] was used. The simulations were performed within a periodic simulation cell with a rectangular parallelepiped shape. The film was positioned in the center of the simulation cell, oriented along the smallest square face of the parallelepiped. The sides of this face were long enough to contain a PE chain of 200 monomers, stretched in each square direction. Previous simulations have demonstrated that this designated interfacial area is large enough to accurately predict the density and interfacial properties of molten polyethylene at temperatures between 373.15 K and 673.15 K [25,26]. The initial conformation of the chains formed a film that completely covered the face of the simulation cell and evolved into a cohesive, self-standing film. The periodicity along the normal direction resulted in a central film that is periodically repeated along the normal axis.

A molten PE film in contact with 938 to 7938 ethane molecules was placed in a rectangular parallelepiped simulation cell with dimensions *A* = 14.5 × 14.5 nm^2^ and *L_z_* = 48.4 nm. This setup allowed us to simulate a thin layer of polyethylene in contact with different ethane gas phase densities (*ρ_et,g_*) ranging from 4.10 × 10^−3^ to 7.97 × 10^−2^ g/mL at 373.15 K. Additional simulations were conducted by progressively reducing *L_z_* to 12.4 nm while keeping the number of ethane molecules constant (7938), which allowed us to reach *ρ_et,g_* values of 1.36 × 10^−1^ g/mL. The reduction of *L_z_* was performed in stages, at a rate of 0.2 nm every 100 ps, with remapping of the coordinates of the entire system, which prevented the system from losing stability. The varying *L_z_* values ensured sufficient separation between the central layer and its periodic images to render inter-layer interactions negligible. The systems were also simulated at 298.15 K and 448.15 K starting from equilibrated configurations at 373.15 K and changing the temperature of the system in several steps. The ethane–PE systems were simulated using the NVT ensemble, where the number of particles and the volume of the simulation cell were kept constant while the temperature fluctuated around a constant value, for which the Nosé thermostat [34] was used with a time step of 1 fs.

The inter-chain interactions (Lennard–Jones potential) and those due to intra-chain vibrations (bond distance, bond angles, dihedral angles, and 1–5 interactions) were calculated using the Transferable Potentials for Phase Equilibria (TraPPE) force field [35]. Graphical representations of the inter- and intra-chain interactions calculated with the Lennard–Jones potential are depicted in Figure 1. This force field uses sites that represent the different monomers along with the Lennard–Jones potential to calculate interaction forces between sites of different chains and also for interactions between sites of the same chain but separated by more than four bonds. Coarse-grained force fields have been widely used to describe the structural properties of polymers [36,37], but the resolution of the components of the pressure profiles $P_{\alpha \beta }(z)$, αβ={xx, yy, zz}, is in the range of a few Å [38], which is difficult to resolve when the interaction sites have diameters of several Å. Bond distance and angle vibrations were calculated using harmonic potentials, while dihedral angle vibrations were calculated using a cosine function [39]. Interactions between sites of different chains were calculated using a large cutoff radius, 7.5 times the value of *σ*_CH3–CH3_. The parameter *σ*_CH3−CH3_ is a Lennard–Jones potential parameter specific for interactions between two terminal –CH_3_ groups of two different chains. This radius was wide enough to consider all significant inter-chain interactions, reproduce experimental interfacial properties (tension), and avoid the use of long-range corrections during and after simulations in systems fully described only with Lennard–Jones potentials [27,40] or even in more complex systems with partial electric charges (ionic) [41,42]. The TraPPE potential has been used to simulate C_50_ and C_192_ PE chains, and these simulations have reproduced their experimental properties, such as melting temperature [43] and glass transition temperature [44]. At high pressures and temperatures, the effective interactions of the TraPPE potential, reproduces well the volumetric properties of supercritical methane at 298 K and 1000 MPa [45].

The density profiles, *ρ(z)*, and the components of $P_{\alpha \beta }(z)$, were calculated using sub-cells with a thickness of 0.1 Å along the normal direction. The pressure tensor components were calculated using the Harasima contours [47], a function available in LAMMPS [33], which arbitrarily distributes the tension of two interacting sites between the two sub-cells in the normal direction that originate the interactions. Although local pressure lacks a unique definition, the Harasima contours accurately describe interfacial pressures in flat interfaces for systems at thermodynamic equilibrium [38]. From the total pressure profiles, The interfacial tension, *γ*, was calculated using its mechanical definition:(1)$$\gamma =\frac{1}{2}\int_{-\infty }^{\infty }\left\{P_{zz}(z)-0.5\left[P_{xx}(z)+P_{yy}(z)\right]\right\}dz$$
where $P_{zz}(z)$ is the component in the normal direction, while the other two are the components lateral to the surfaces that form the interfaces. Equation (1) considers the presence of two interfaces.

## 3. Results and Discussion

Figure 2 shows the equilibrium conformations of the molten PE films in contact with supercritical ethane at 373.15 K. The visualizations correspond to the low and high *ρ_et,g_* studied in this work. These conformations were taken after 10 ns of initial simulation to reach equilibrium conformations. The thin polyethylene layer widens as *ρ_et,g_* increases, a phenomenon that likely reduces the PE film density. At high *ρ_et,g_* ethane molecules accumulate at the polymer/gas interfaces at densities greater than those in the center of the film.

### 3.1. Density Profiles

The *ρ(z)* for each component (PE and ethane) and the sum of both (total) along the axis normal to the interfacial surfaces, were calculated over a period of 5 ns after an initial 10 ns of equilibration. The density profiles for the 94 PE chains at 373.15 K are shown in Figure 3, while the density profiles for ethane are shown in Figure 4. To show the data dispersion, Figure 3 and Figure 4 also show the *ρ(z)* averaged every 0.1 ns and the overall average over 5 ns. The PE profiles (Figure 3) generally show low data dispersion, with maximum differences in the center of the layer of up to 0.015 g/mL, 0.020 g/mL, and 0.030 g/mL for the system without ethane molecules, and the systems with ethane at *ρ_et,g_* of 4.10 × 10^−3^ g/mL and 1.36 × 10^−1^ g/mL, respectively. Figure 3 also shows the density profile of the PE chains before coming into contact with supercritical ethane at the same temperature. In previous reports it has been shown that the maximum value for the density of the film is almost independent of the width of the film for systems dominated by van der Waals forces [25,28]. The *ρ(z)* show that the molten film widens in the presence of ethane, and that the film’s central density decreases as *ρ_et,g_* increases, similar to a surfactant effect. Similar effects have been found in phase equilibrium systems of binary mixtures of long-chain alkanes like *n*-decane in thermodynamic equilibrium with short-chain alkanes like methane and ethane [48,49,50].

The film thickness formed only by PE chains, was defined as the separation between interfacial points where the PE density (Figure 3) was half its maximum central value. Using this criterion, the analysis shows that the PE film did not widen when it was put in contact with ethane at the lowest *ρ_et,g_* studied (4.10 × 10^−3^ g/mL). However, it did widen by approximately 30% when it was put in contact with the gas phase at the highest *ρ_et,g_* studied (1.36 × 10^−1^ g/mL). Considering also only the PE chains, the maximum density at the center of the layer is reduced by less than 1% for the lowest *ρ_et,g_* studied and approximately 22% for the highest *ρ_et,g_* studied.

The *ρ(z)* corresponding to ethane (Figure 4) shows a pronounced concentration at the interfaces, which manifests as adsorption peaks near the surfaces of the PE film. Similar adsorption peaks have been reported in previous studies of low molecular weight gases in liquid/vapor equilibrium with long alkane chains [51,52,53], and are also present in many systems with polar and non-polar components [54]. The *ρ(z)* show that as *ρ_et,g_* increases, the maximum density of the ethane adsorption peaks, *ρ_et,m_*, also grows, and can reach values as high as approximately ⅓ of the density values of the PE chains in the center of the PE film. The widening of the polyethylene film as *ρ_et,g_* increases is also reflected in the separation of the ethane adsorption peaks at the interfaces. This separation widens from approximately 2.82 nm (*ρ_et,g_* = 4.10 × 10^−3^ g/mL) to approximately 4.31 nm (*ρ_et,g_* = 1.36 × 10^−1^ g/mL), representing an increase of approximately 52%. In the center of the film, the average ethane density, *ρ_et,c_*, reaches values similar to *ρ_et,g_* for the system with the lowest *ρ_et,g_* studied at 373.15 K. In the system with the highest *ρ_et,g_* studied, *ρ_et,c_* decreases to approximately 2/3 of the *ρ_et,g_* value, which could indicate that a saturation process for ethane adsorption is in progress in the center of the PE layer, but not in the adsorption peaks, which continue to grow as *ρ_et,g_* increases.

The total *ρ(z)*, including both components of the system, are shown in Figure 5. Compared to the PE profiles, a less pronounced decrease in density is observed in the center of the layers. For the system with *ρ_et,g_* = 4.10 × 10^−3^ g/mL, there is no density reduction. However, for the system with *ρ_et,g_* = 1.36 × 10^−1^ g/mL, the reduction is only 8.8%, which is probably due to the solubility of ethane molecules in the PE film. The widening of the layer is due to the progressive widening of the adsorbed ethane layers at the interfaces, but is also due to a reduction in the PE chains density. For the system with the lowest *ρ_et,g_* studied, the widening is insignificant, but for the system with the highest *ρ_et,g_* studied, the widening is significant, approximately 60%, estimated from the total *ρ(z)* positions where the density begins to deviate from the average *ρ_et,g_*.

### 3.2. Pressure Profiles

The $P_{\alpha \beta }(z)$ in the normal (zz) and lateral (xx and yy) directions were calculated for all systems, and used to estimate the average pressure in the ethane gas phase, $P_{T,g}$, and also used to calculate the interfacial tension, *γ*. The pressure component profiles from the simulations at 373.15 K, at the limit values of the *ρ_et,g_* range studied, are shown in Figure 6. These profiles are compared with those obtained for the pristine PE film, without being exposed to ethane molecules. The effect of ethane in the PE film reduces the magnitude of the peaks and valleys of the 3 pressure components. The presence of ethane makes the PE film less cohesive. This might seem counterintuitive, as one might expect a small molecule like ethane to fill the natural pores in the films [25] and increase stability. For comparison, interstitial carbon atoms in steel form a solid solution, creating a harder but more fragile system [55]. However, in our simulations, ethane appears to have the opposite effect, suggesting its main influence is at the interfaces, making the film more fragile.

The $P_{\alpha \beta }(z)$ show a behavior corresponding to a metastable system. At the ethane gas phase, the pressure is in equilibrium and is isotropic. Its 3 components have equal average values, which is not the case in the center of the PE layer, where one would expect the development of an isotropic phase similar to a liquid. However, this isotropy is not achieved. The lateral pressure components $P_{xx}(z)$ and $P_{yy}(z)$ are equal and indicate cohesive conformations, but they differ from the normal component $P_{zz}(z)$, which has positive, non-cohesive values. This pressure difference causes the system not to be in mechanical equilibrium. The mechanical imbalance will eventually cause these systems to transform into more stable systems, which could occur through the film breaking and the formation of an aerosol [27,56,57]. The stability of these molten films depends on many external factors. Although classical MD simulations without external perturbations have shown long periods of stability, up to 0.1 μs, with potentials based on interaction sites of –CH_2_– and –CH_3_ functional groups [25], and also with mesoscale simulations [58].

The $P_{\alpha \beta }(z)$ in the interfacial region show more cohesive states in the lateral directions than in the normal direction, which is natural and gives rise to the development of *γ*. This difference is a result of the inhomogeneity of the system in the normal direction. The studied self-standing PE film is basically interfacial; there is no region identifiable as a bulk liquid, in which the three pressure components should develop with isotropic and positive values to maintain mechanical equilibrium with the ethane gas phase. A comparison of the change in pressures between the minimum value at the interfaces and its corresponding maximum in the center of the layer, indicates that the magnitude of the change is the same in the three components. Such large changes in the normal direction are not expected in wider layers [41], which probably indicates interference of the opposite interfaces. In wider layers, the expected deviations from a constant $P_{zz}(z)$ are considerably lower than those observed in the lateral directions, and the small deviations are due to the way the interactions in the $P_{zz}(z)$ are accumulated in the Harasima profiles.

Based on the $P_{\alpha \beta }(z)$ obtained, the profiles of the function $P_{zz}(z)-\;0.5[P_{xx}\left(z\right)+P_{yy}(z)]$ were calculated. These profiles, which are shown in Figure 6d, once integrated, are used to calculate *γ* (Equation (1)). Compared to the profiles of the pristine PE film without ethane, the presence of ethane causes the maximums of the interfacial peaks to reduce their magnitude. Furthermore, when *ρ_et,g_* is high enough, non-cohesive regions appear in the outermost part of the interfaces, which could be investigated in the future using a code implemented in such a way that it allows for the separate calculation of the contributions of each component to the profiles, and in this way determine if these non-cohesive regions are due to the PE chains or the ethane molecules. This is not possible today with the MD implementation used [33]. Figure 6d also shows that the system is composed of two overlapping interfacial regions, with no central bulk liquid exhibiting isotropic pressures. This is because the influence tails of the two interfacial peaks overlap in the center of the layer. The snapshot of Figure 6e shows the limits of the two interfaces for the system at 373.15 K and the smallest *ρ_et,g_* studied in this work. Each interface is composed of several sublayers of chains in the normal direction, and many more layers are needed to develop a liquid phase.

Figure 7 shows the profiles of the main contributions (van der Waals inter-chain interactions + 1–5 intra-chain interactions, bond distance vibrations, and bond angle vibrations) to the function $P_{zz}(z)\;-\;0.5[P_{xx}(z)+P_{yy}(z)]$, along with the total profile and the profiles for the pristine PE film. In the pristine film, it is shown that the vibrations contribute mainly in the outermost parts of the interfaces, while the inter-chain interactions (van der Waals forces) are divided between regions that have positive and negative contributions. The negative contributions, like the vibrations, are located in the outermost part of the interfaces, followed by the positive regions in the center of the film. The positive contributions due to vibrations are greater than the effect of the negative contributions due to inter-chain interactions in the outermost regions of the interfaces. The $P_{\alpha \beta }(z)$ due to bond distance and bond angle vibrations do show an isotropic region that develops in an internal sub-layer located at the center of the film, with a thickness of ~ 1.4 nm. The thickness of this isotropic region increases slightly at higher *ρ_et,g_*. The contributions due to inter-chain interactions do not show any isotropic region at the center of the film.

### 3.3. Pressure–Density of Supercritical Gas Ethane

$P_{T,g}$ represents the average value of the three $P_{\alpha \beta }(z)$ components in the homogeneous regions, away from the film and its interfaces, where the density varies around a constant value. Figure 8 shows the values obtained for $P_{T,g}$ along with its corresponding *ρ_et,g_*, as well as the corresponding values for *ρ_et,c_* and *ρ_et,m_*. The results are also shown at temperatures of 298.15 K and 448.15 K, to have a more global view of the behavior of these systems. Previous work with pristine PE films had shown that at higher temperatures, a greater number of chains was needed to keep the film stable for long simulation periods (100 ns). The systems at 448.15 K remained stable, during the short simulation period studied (15 ns), probably because simulations were started from configurations with the ethane molecules, which gives some sort of strength to the film.

Figure 8 also shows experimental isotherms of the $P_{T,g}$–*ρ_et,g_* relationship for pure supercritical ethane at the 3 studied temperatures [59]. The used potential reproduces the experimental ethane $P_{T,g}$–*ρ_et,g_* relationship well. There are slight over-predictions at high *ρ_et,g_*, with maximum errors of 6.25% within the $P_{T,g}$–*ρ_et,g_* region studied. The behavior between these variables is monotonically increasing, although at 298.15 K, and slightly higher *ρ_et,g_*, the experimental $P_{T,g}$ increases exponentially, or viewed another way, there is a saturation of $P_{T,g}$ at higher *ρ_et,g_*. *ρ_et,m_* and *ρ_et,c_* grow with *ρ_et,g_*, although *ρ_et,m_* grows asymptotically and *ρ_et,c_* grows linearly, which could indicate that the phenomenon that is entering a saturation behavior in this studied density range is the phenomenon of ethane adsorption at the interfaces, but this value does not represent the average density at the peaks, therefore is not conclusive. The solubility in the center of the film has not really started a saturation process because it is growing linearly. *ρ_et,m_* presents magnitudes up to 2.4 times greater than *ρ_et,c_*, although it is not clear if the isotherms could connect at higher *ρ_et,g_*. At 298.15 K, the densities of *ρ_et,c_* are greater than *ρ_et,g_*, but at higher temperatures, there is an inversion in this relationship that also becomes more pronounced as the temperature increases, which is probably related to the melting temperature for pure PE, which occurs in this range of temperatures for ultra-thin layers of polyethylene, which have melting temperatures lower than bulk polyethylene [60].

### 3.4. Interfacial Tension

The integration of the profiles of the function $P_{zz}(z)\;-\;0.5[P_{xx}\left(z\right)+P_{yy}\left(z\right)]$, were used to calculate *γ* (Equation (1)). For this, the total values of $P_{\alpha \beta }(z)$ were used. The *γ* values obtained are reported in Figure 9 for the 3 temperatures studied. As $P_{T,g}$ increases, *γ* decreases, and this negative growth fits well to the behavior of an exponential decay function. It has been previously reported that this potential reproduces the experimental *γ* of pure PE films [26], finding an independence of *γ* with the thickness of the PE film. Therefore, *γ* is independent of whether the film is metastable or stable. Ethane has the effect of a surfactant on the metastable molten PE film, and its effect is more pronounced at low temperatures. At 298.15 K, *γ* drops on average by 17.5% with each MPa increase in $P_{T,g}$, while at 448.15 K the decrease is ~4.8% for each MPa.

The three isotherms reach similar *γ* values at high $P_{T,g}$, while at the lowest $P_{T,g}$, the curves behave as expected; *γ* values at zero $P_{T,g}$ depend only on the temperature of the pristine PE film. The literature has widely documented that in liquid/vapor thermodynamic equilibrium systems, *γ* is minimally affected by pressure but greatly influenced by liquid density [61]. Given that at the highest $P_{T,g}$ studied, the only difference between the 3 systems is the temperature (the cell volume and the overall density are the same), then a more fair comparison can be made using the change they underwent from the pristine systems at $P_{T,g}$ = 0 MPa. Compared to the pristine PE films, the films in contact with methane at high $P_{T,g}$ take lower total densities of 0.768 g/mL (298.15 K), 0.7383 g/mL (373.15 K), and 0.694 g/mL (448.15 K), while the corresponding percentage reductions were more or less constant, of 9.49%, 8.99%, and 10.23%, respectively. This probably indicates that the dependence of *γ* should probably be related to the change in the magnitude and not to the total value of the film PE density, as commonly occurs in bulk phases under thermodynamic equilibrium.

### 3.5. Interfacial and Soluble Load

The amount of ethane the PE film can accumulate, $q_{T}$, sum of both what is dissolved in the center of the film, $q_{S}$, and what is adsorbed at the interfaces, $q_{I}$, are shown in Figure 10 for all the temperatures studied. $q_{T}$ at all temperatures studied grows exponentially with $P_{T,g}$, and the growth rate decreases as the temperature increases. Experimental studies [19] and numerical calculations using SAFT [20,22,23], have found a similar behavior for $q_{S}$ in systems composed of different forms of bulk PE, exposed to supercritical gases like *n*-pentane, propane, ethene, propene, but-1-ene, and ethylene. These studies were carried out with bulk PE where the interfacial adsorption part plays an insignificant role. However, for PE in film form, interfacial adsorption is very significant, as can be seen in the ρ(z) of ethane (Figure 4). No reported studies were found in the literature of ethane adsorbed in any form of PE.

Studies with PE films at 298.15 K have also found that the $q_{T}-P_{T,g}$ trend reported in this work is the same when the PE film was in contact with supercritical ethylene, but the trend changed to an asymptotic behavior when the PE was exposed to a lighter gas like supercritical methane [12]; this has also been reported for combinations of heavier supercritical alkanes in bulk PE [22]. The experimental results reported in the literature with PE films [12,24] cannot be directly compared with the results of this work, since in the experiments the PE films are pre-treated with *n*-heptane and the thickness of the experimental films was not reported, which can affect the degree of adsorption and solubility of ethane. The values obtained in this work using pristine, untreated PE films at the stability limit thickness accumulated up to 40 times what was observed experimentally at 4 MPa using ethylene and polyethylene films pre-treated with *n*-heptane and without knowledge of their thickness [12]. Results for $q_{T}$ obtained at 298.15 K fall between those obtained experimentally adsorbing CO_2_ and H_2_ on HDPE films at 0.3 MPa and 303.15 K [10].

The solubility of ethane in the PE films was estimated by considering that there is a region in the center of the liquid that exhibits mechanical stability, at least in terms of the vibrations of the chain sites (Figure 7), but not in their inter-chain interactions due to van der Waals forces. $q_{S}$ reaches values between 2.94 g and 74.8 g per 1000 g of PE for the systems at 373.15 K and $P_{T,g}$ of 0.29 MPa and 9.65 MPa, respectively. These values show that $q_{T}$ varies between 4.8 and 8.6 times the $q_{S}$ value when $P_{T,g}$ varies between 0.29 MPa and 9.65 MPa, at 373.15 K.

## 4. Conclusions

Classical Molecular Dynamics simulations were carried out to understand how supercritical ethane molecules accumulate, distinguishing between the ethane that dissolves and that which accumulates at the polymer/gas interface of a polyethylene film at its stability limit thickness. Results at temperatures of 298.15 K, 373.15 K, and 448.15 K, and pressures up to 13.17 MPa, show that the total ethane load grows exponentially as the pressure of the ethane in the gas phase increases. This is similar to what has been found in experiments with polyethylene in film and bulk form in equilibrium with light gases in supercritical state. No experimental studies on ethane were found that would allow us to compare our results, not even with bulk polyethylene. The accumulation capacity of ethane in these two regions decreases as the system temperature increases. An estimation of the increased adsorption due to interfacial ethane adsorption can be between 4.8 and 8.6 times the solubility that ethane naturally exhibits when absorbed in bulk polyethylene at 373.15 K and pressures up to 9.65 MPa. The high adsorption at the interfaces, compared to its solubility in the bulk, suggests the development of arrangements made of multiple polyethylene films as a promising strategy for developing adsorbents for problematic light gases in various industrial and environmental processes.

The inclusion of ethane on polyethylene films, which have thicknesses at their mechanical limit of stability, does not change how metastable they are. While contributions to surface tension from chain vibrations exhibit mechanical stability and the systems remain stable for up to 15 ns, they are not mechanically stable with respect to inter-chain van der Waals interactions. This is likely because these systems are mainly composed of two interfaces whose influence tails overlap in the center of the film. Thicker films are necessary to prevent the interfaces from interacting and to make the films mechanically stable. The interfacial tension exhibits an exponential decay with the pressure of the gas phase, and the rate of decay slows as temperature increases.

The solubility of ethane in the polyethylene film changes with the temperature. At 298.15 K, the density of dissolved ethane is greater than that of the gas phase. This relationship reverses at 373.15 K and becomes more pronounced with increasing temperature, which is probably related to the occurrence of the melting temperature of polyethylene films in this range of temperatures. It will be interesting to investigate if both events occur at the same temperature.

Future work will determine if the limit of mechanical stability is reduced in the presence of ethane, as well as how selective are these films for the adsorption capacity of certain problematic light gases. On the other hand, experimental verification of the results of this work is essential but complex, because it requires the use of state-of-the-art technology for the production of ultra-thin polyethylene films.

## Figures and Tables

**Figure 1 polymers-17-02751-f001:**
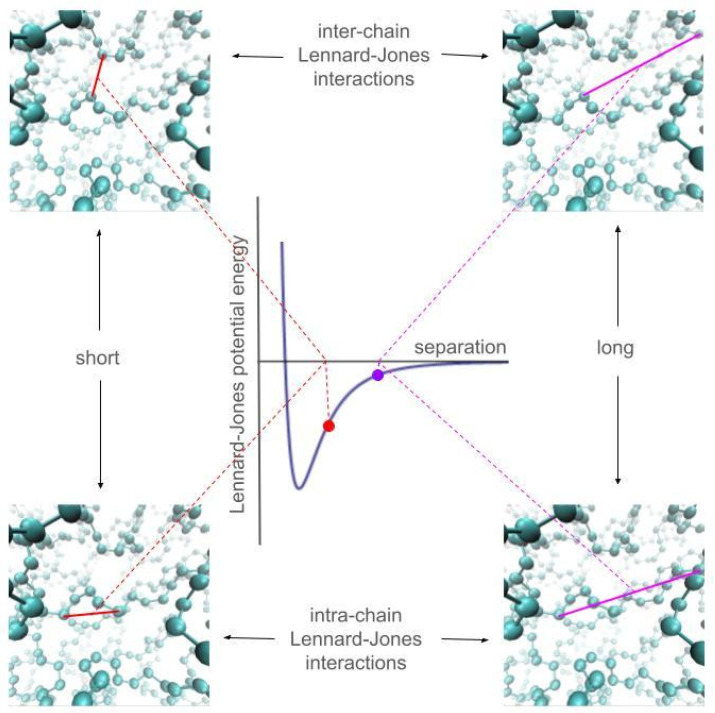
Graphical representations of the inter- and intra-chain interactions for polyethylene chains. Purple and red spheres represent –CH_2_– and –CH_3_ functional groups in polyethylene chains, respectively. The Lennard–Jones potential was employed to calculate these interactions; long separations are not accounted for unless the separation between interaction sites is lower than the cutoff radius. At the cutoff radius interaction energies are insignificant, as the separation between interaction sites becomes shorter, it produces larger cohesive energies. Beyond the minima in the potential energy curve, interactions became repulsive and incohesive. The figures were generated with the VMD visualizer [46].

**Figure 2 polymers-17-02751-f002:**
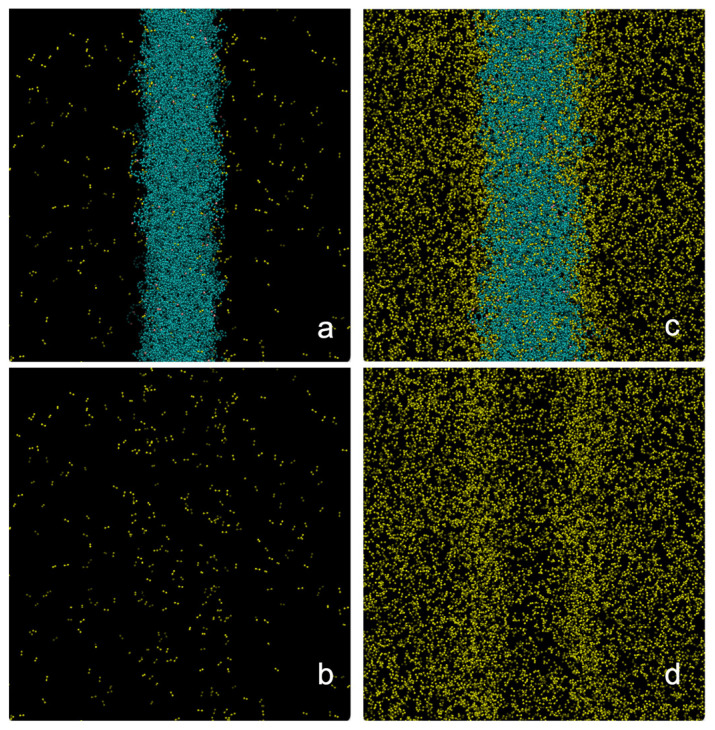
Equilibrium configurations of a polyethylene film at 373.15 K in equilibrium with supercritical ethane at (**a**) 4.10 × 10^−3^ g/mL and (**c**) 1.36 × 10^−1^ g/mL. Ethane molecules are shown in yellow spheres and the polyethylene film in blue spheres. Panels (**b**,**d**) show the same systems with the polyethylene chains removed for clarity. The figures were generated with the VMD visualizer [46].

**Figure 3 polymers-17-02751-f003:**
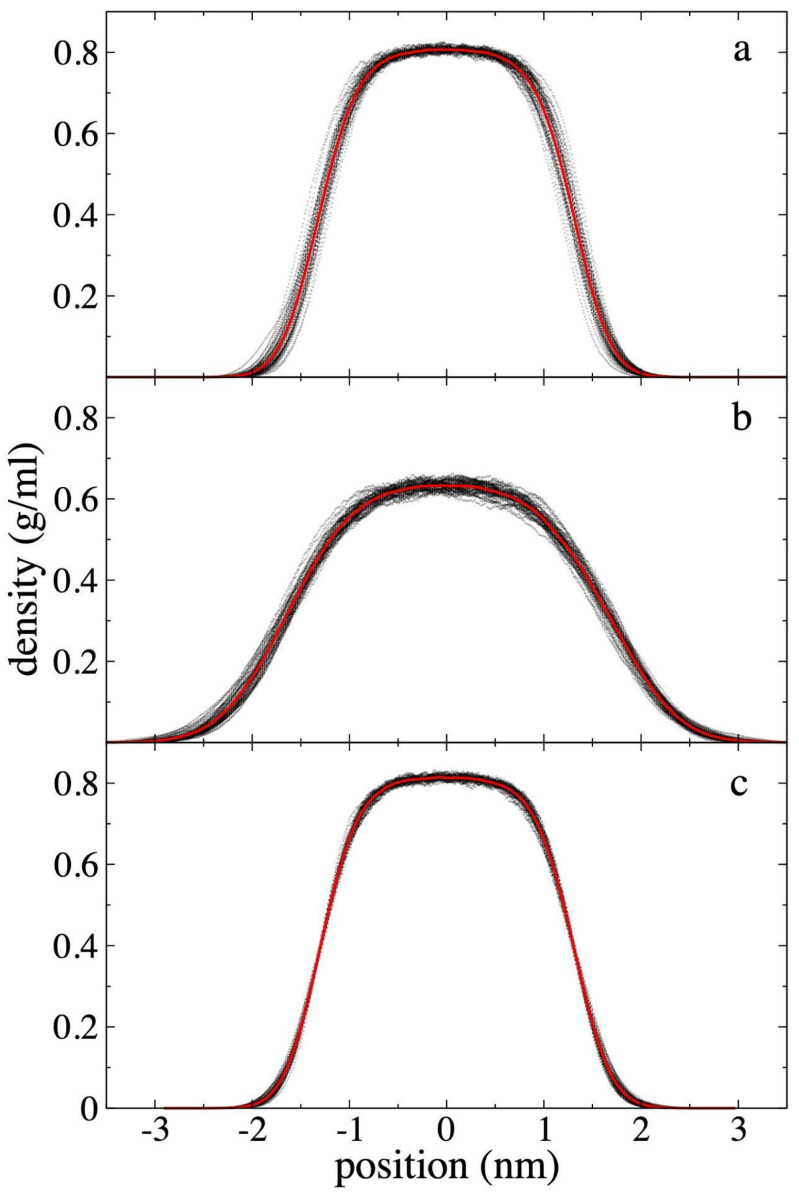
Polyethylene density profiles at 373.15 K. Profiles correspond to polyethylene films in equilibrium with supercritical ethane having gas densities of (**a**) 4.10 × 10^−3^ g/mL and (**b**) 1.36 × 10^−1^ g/mL. (**c**) shows the density profiles for a pristine polyethylene film. Profiles averaged every 0.1 ns are shown with black dots, and the overall average over a 5 ns period in red dots.

**Figure 4 polymers-17-02751-f004:**
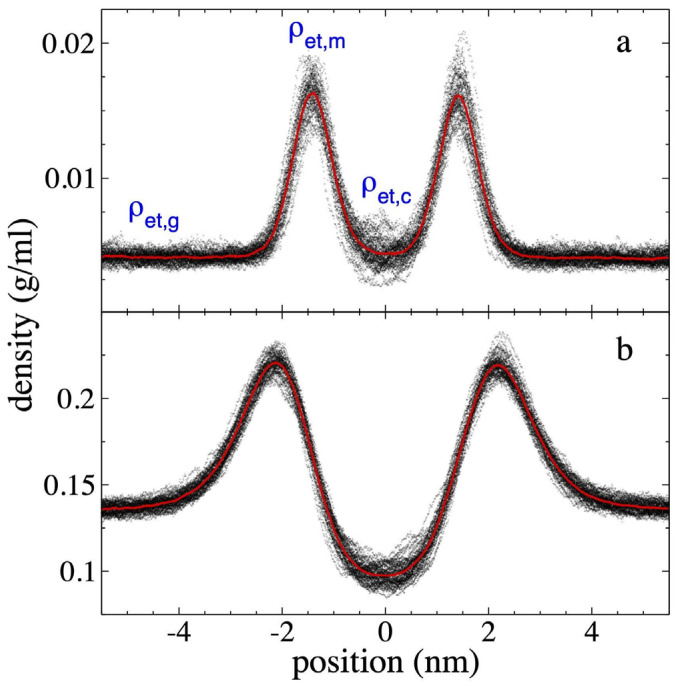
Ethane density profiles at 373.15 K. Profiles correspond to polyethylene films in equilibrium with supercritical ethane having gas densities of (**a**) 4.10 × 10^−3^ g/mL and (**b**) 1.36 × 10^−1^ g/mL. Profiles averaged every 0.1 ns are shown with black dots, and the overall average over a 5 ns period in red dots.

**Figure 5 polymers-17-02751-f005:**
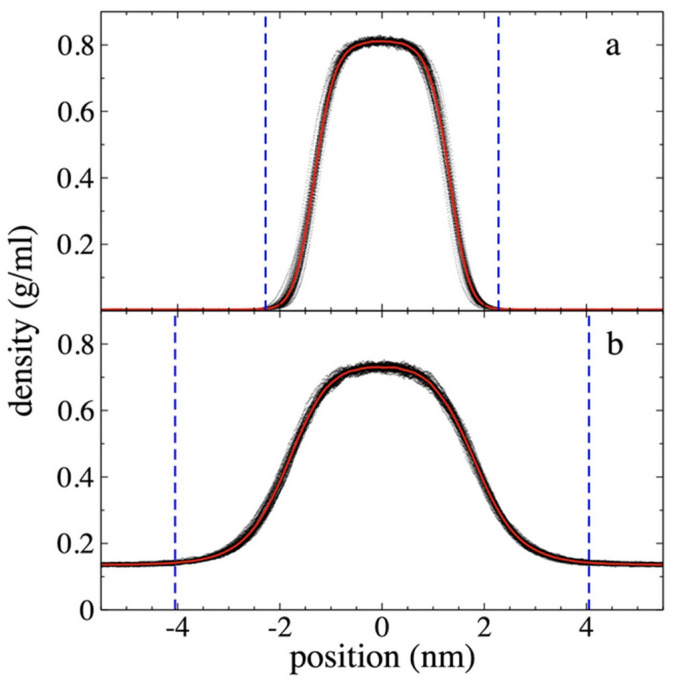
Total density profiles at 373.15 K. Profiles correspond to polyethylene films in equilibrium with supercritical ethane having gas densities of (**a**) 4.10 × 10^−3^ g/mL and (**b**) 1.36 × 10^−1^ g/mL. Profiles averaged every 0.1 ns are shown with black dots, and the overall average over a 5 ns period in red dots. Blue lines represent the limits of the polyethylene film with the adsorbed interfacial ethane layers.

**Figure 6 polymers-17-02751-f006:**
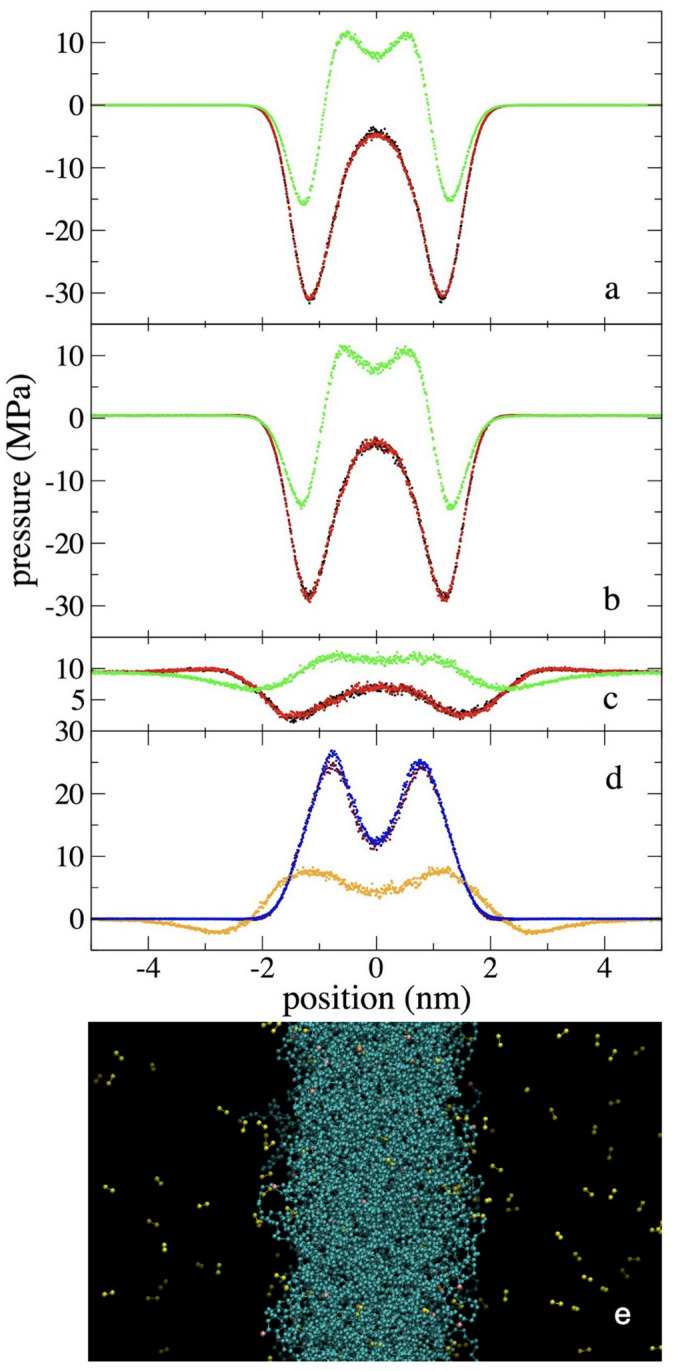
Lateral and normal pressures profiles at 373.15 K, for systems representing (**a**) the pristine polyethylene films, and the film in equilibrium with supercritical ethane having gas densities of (**b**) 4.10 × 10^−3^ g/mL and (**c**) 1.36 × 10^−1^ g/mL. The two lateral pressures (black and red) and the normal pressure (green) are shown. (**d**) Profiles of the function $P_{zz}(z)\;-\;0.5[P_{xx}\left(z\right)+P_{yy}\left(z\right)]$ corresponding to systems (**a**) blue dots, (**b**) brown dots, and (**c**) orange dots. (**e**) Configuration of a polyethylene film at 373.15 K in equilibrium with supercritical ethane at 4.10 × 10^−3^ g/mL; red spheres represent methyl and blue methylene groups in polyethylene, while yellow spheres represent methyl group in ethane. The figure was generated with the VMD visualizer [46].

**Figure 7 polymers-17-02751-f007:**
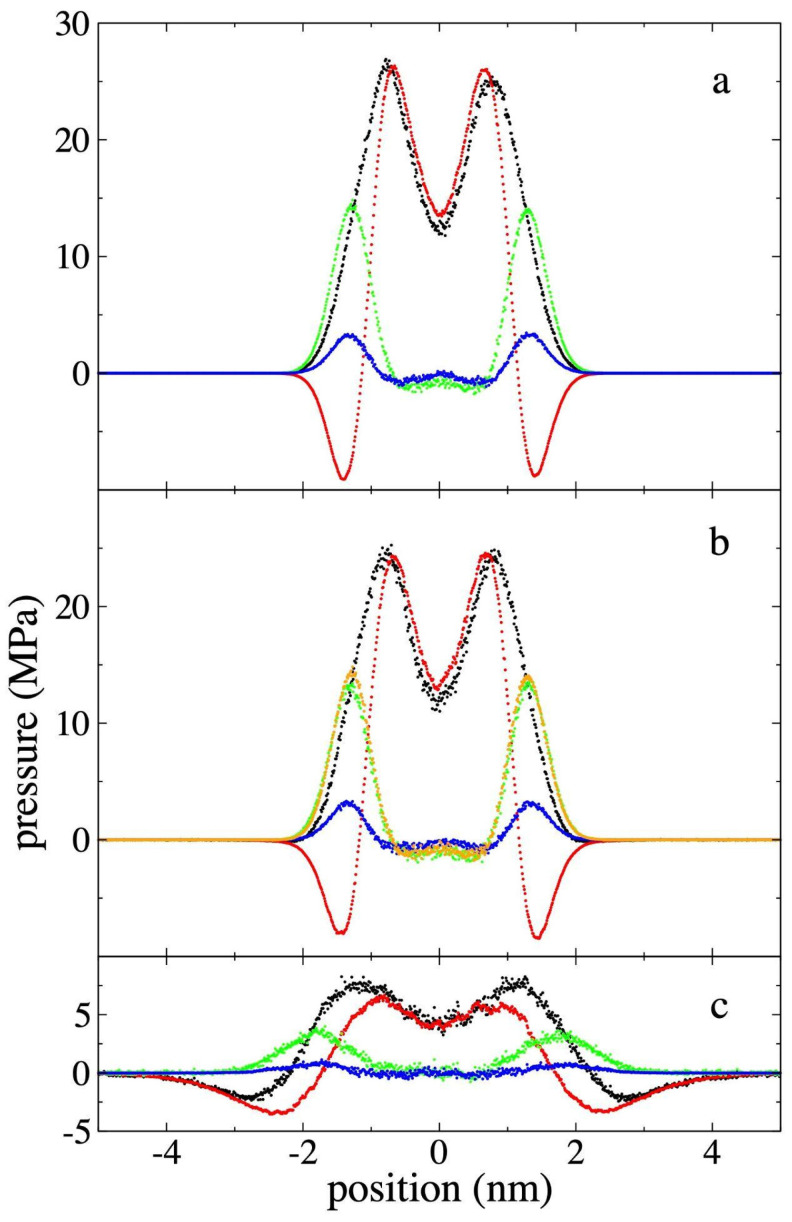
Profiles of the main contributions to the function $P_{zz}(z)\;-\;0.5[P_{xx}(z)+P_{yy}(z)]$ at 373.15 K, for systems representing (**a**) the pristine polyethylene films, and the film in equilibrium with supercritical ethane having gas densities of (**b**) 4.10 × 10^−3^ g/mL and (**c**) 1.36 × 10^−1^ g/mL. The sum of all contributions are plotted with black dots, the contributions of inter-chain interactions with red dots, bond vibrations with green dots, and angle vibrations with blue dots. The contributions due to bond vibrations from (**a**) are also shown in (**b**) with orange dots for comparison.

**Figure 8 polymers-17-02751-f008:**
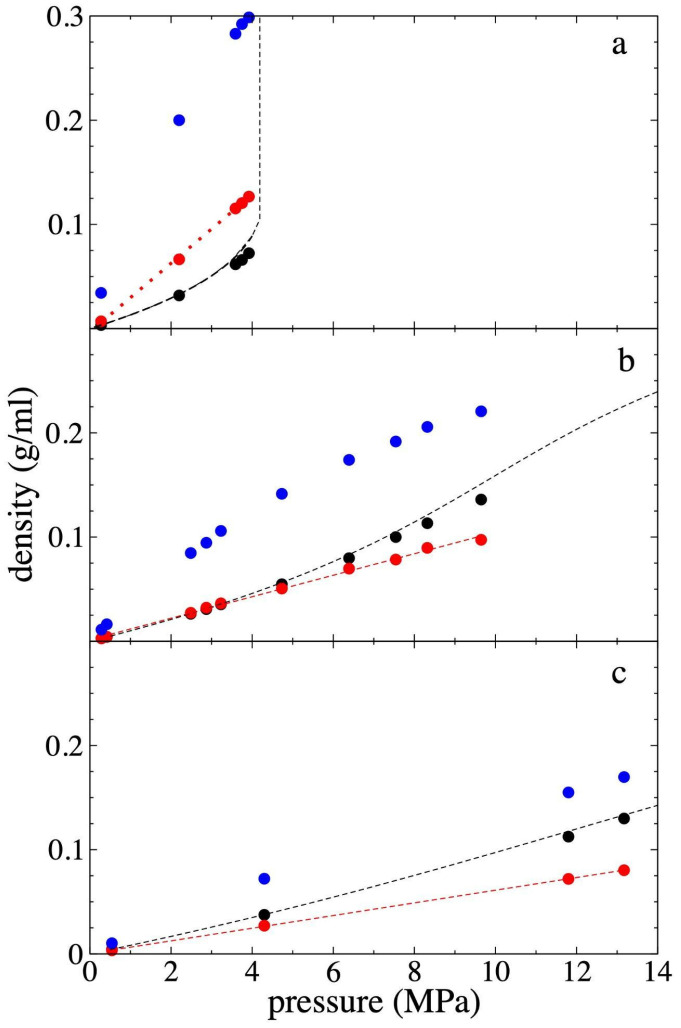
Gas density–pressure isotherms for supercritical ethane in equilibrium with the polyethylene film at (**a**) 298.15 K, (**b**) 373.15 K and (**c**) 448.15 K. Data points represent the average density of bulk ethane gas (black circles), maximum adsorption peak density (blue circles), and average ethane density at the film’s center (red circles). Solid black lines indicate experimental data [59]. Red lines represent linear regressions.

**Figure 9 polymers-17-02751-f009:**
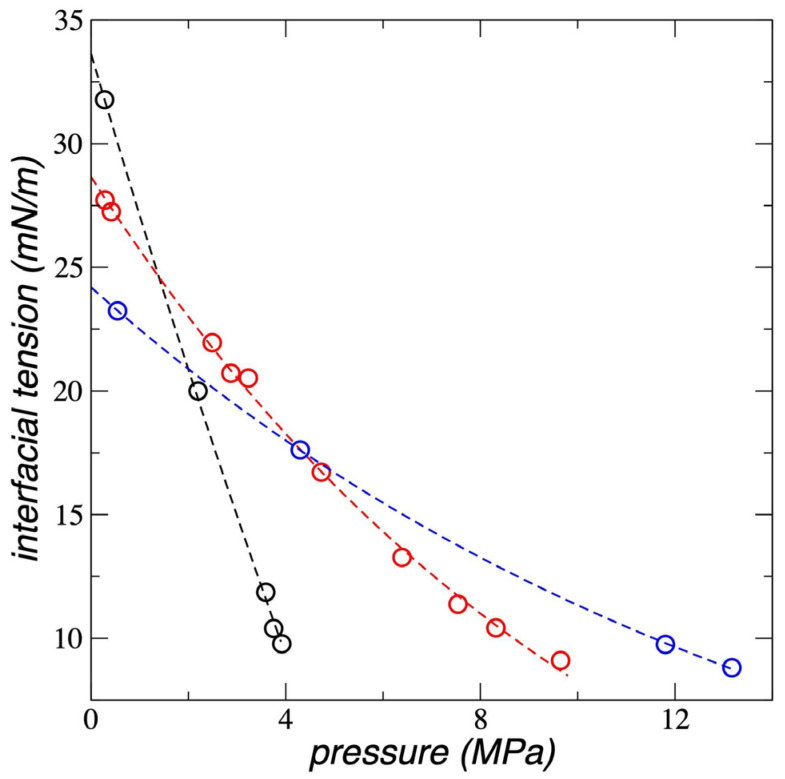
Interfacial tension as a function of the bulk pressure of the gas ethane, for systems representing polyethylene films in equilibrium with supercritical ethane, at temperatures of 298.15 K (black circles), 373.15 K (red circles) and 448.15 K (blue circles). The corresponding lines represent the fitting to an exponential decay function.

**Figure 10 polymers-17-02751-f010:**
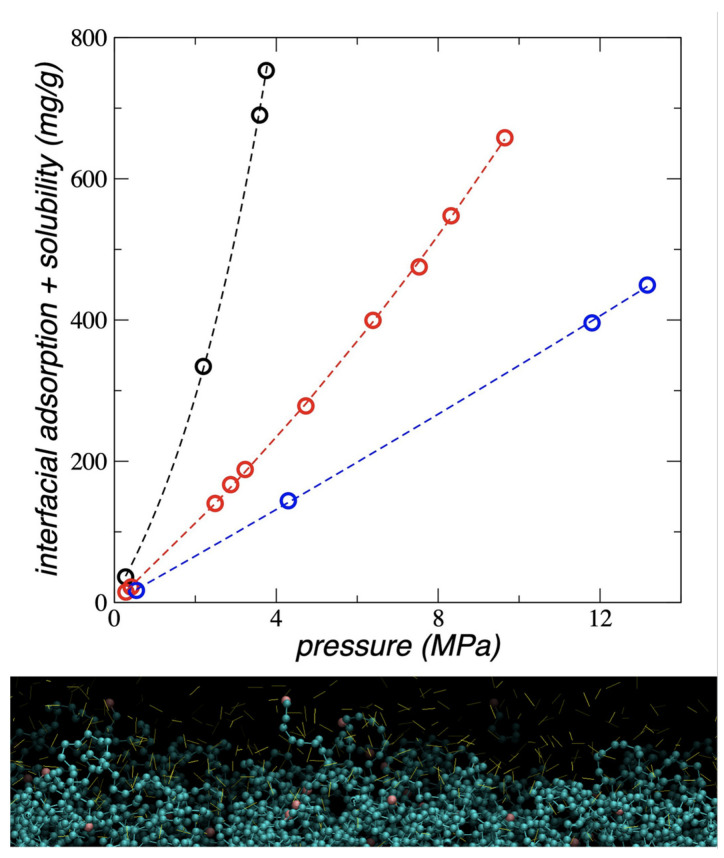
Top: total load as a function of the bulk gas pressure for polyethylene films in equilibrium with supercritical ethane, at 298.15 K (black circles), 373.15 K (red circles) and 448.15 K (blue circles). The corresponding lines represent the fitting to exponential growth functions. Bottom: fractions of polyethylene chains desorbs and accumulate ethane molecules at the interfaces. The image shows a snapshot of the configuration of a polyethylene film at 373.15 K in equilibrium with supercritical ethane at 4.72 × 10^−2^ g/mL; red spheres represent methyl and blue methylene groups in polyethylene, while yellow lines represent ethane molecules. The figure was generated with the VMD visualizer [46].

## Data Availability

The original contributions presented in this study are included in the article. Further inquiries can be directed to the corresponding author(s).

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
