# Peer review of "High Interfacial Adsorption of Light Gases on Nano-Thin Molten Polyethylene Films"

_polymers, 2025, doi:10.3390/polym17202751_

Round 1

Reviewer 1 Report

Comments and Suggestions for Authors

This paper investigates the adsorption and solubility of supercritical ethane on pristine nano-thin molten polyethylene (PE) films using classical molecular dynamics (MD) simulations. The authors focus on metastable molten PE films at their mechanical stability limit and analyze the preferential distribution of ethane between interfacial and bulk regions across a range of temperatures (298.15–448.15 K) and pressures (0.28–13.17 MPa). The manuscript is generally well written, and the findings are relevant to the field. However, before acceptance, the following minor revisions should be considered:

  1. The current title is long and somewhat cumbersome. A more concise and engaging formulation would improve clarity and readability.

  2. Informal expressions such as “We investigate…” or “Here, we study…” should be reformulated into a more formal scientific tone, e.g., “It was investigated…” or “This study examines…”.

  3. Although no experimental studies on ethane in PE films are available, the authors should strengthen their discussion by comparing their results with existing data for similar gases such as methane, COâ‚‚, or propane.

  4. The industrial relevance of the findings should be clarified. Are the results applicable to gas separation membranes, packaging films, or environmental adsorption? Highlighting these aspects would significantly enhance the impact of the work.

  5.  Future work should include targeted experiments on ethane sorption in PE thin films with controlled thicknesses to validate the simulation predictions.

Overall, this is a valuable contribution, but addressing the points above would improve its clarity, rigor, and broader significance.

Author Response

We thank the reviewer for his/her valuable comments.

Comment 1: The current title is long and somewhat cumbersome. A more concise and engaging formulation would improve clarity and readability.

Response 1: We eliminated some parts of the title looking to keep the main features of the work.

Comment 2: Informal expressions such as “We investigate…” or “Here, we study…” should be reformulated into a more formal scientific tone, e.g., “It was investigated…” or “This study examines…”

Response 2: We corrected the informal phrases in several parts of the text. They are highlighted in blue.

Comment 3: Although no experimental studies on ethane in PE films are available, the authors should strengthen their discussion by comparing their results with existing data for similar gases such as methane, COâ‚‚, or propane.

Response 3: We added a paragraph (lines 451-453) comparing our results to the experimental work with polyethylene films and CO2 and H2 gases (Ref. 10). At  0.3 MPa and 303.15 K our results fall between those obtained for these two gases, which are in line with the dependence of the solubility with the molecular weight of the gases (Ref. 19 and 20).

Comment 4: The industrial relevance of the findings should be clarified. Are the results applicable to gas separation membranes, packaging films, or environmental adsorption? Highlighting these aspects would significantly enhance the impact of the work.

Response 4: We eliminated the generic first paragraph of the introduction and added a new paragraph (lines 31-44) describing the importance of adsorption materials for ethane adsorption in petrochemical and environmental processes.

Comment 5: Future work should include targeted experiments on ethane sorption in PE thin films with controlled thicknesses to validate the simulation predictions.

Response 5: We added a comment in the Future Work section declaring the importance of determining experimentally the properties studied in this work (lines 498-500).

.We rewrote some parts of the text, to make the text more clear. They are highlight in green.

Reviewer 2 Report

Comments and Suggestions for Authors

This article employs classical Molecular Dynamics simulations to investigate the interfacial adsorption of supercritical ethane on ultrathin molten polyethylene (PE) films under various temperature (298.15-448.15 K) and pressure (0.28-13.17 MPa) conditions. The study analyzes spatial density and pressure profiles of PE films, evaluating ethane’s adsorption and dissolution at the interfaces as well as its influence on the PE film structure.

The following issues should be addressed:

1) The authors employ the TraPPE force field and Lennard-Jones (LJ) interactions to model the interfaces, neglecting electrostatic effects. Can such a force field reliably describe PE-ethane interactions under high-pressure, high-temperature conditions?

2) In the methodology, the representative models should be presented graphically rather than relying solely on textual descriptions.

3) The manuscript does not provide structural details of PE-ethane interactions. These should be included, particularly for statements such as "ultrathin metastable films at the mechanical stability limit are composed of two overlapping interface" and "interfacial tension decreases exponentially with increasing gas pressure and is primarily governed by inter-chain interactions at the interface." Local simulation snapshots or schematic illustrations are necessary to help readers construct a physical picture.

4) The films studied here are pristine, untreated PE. Can the obtained data be fairly compared with experimental films reported in the literature, which were pretreated with n-heptane and had unknown thickness? If not, could the conclusions be overestimating interfacial adsorption?

5) The results show that the density of ethane in the PE center undergoes a reversal with increasing temperature, while interfacial adsorption also decreases with increasing temperature. The authors should provide a reasonable explanation: for example, is this reversal directly related to the melting point, crystallinity, or microstructural characteristics of PE films? Can local structural statistics from simulations support these interpretations?

Author Response

We thank the reviewer for his/her valuable comments.

Comment 1: The authors employ the TraPPE force field and Lennard-Jones (LJ) interactions to model the interfaces, neglecting electrostatic effects. Can such a force field reliably describe PE-ethane interactions under high-pressure, high-temperature conditions?

Response 1: This work only studied volumetric and interfacial properties and the TraPPE forcefield reproduces well these properties. We now included a report from the literature about the use of the TraPPE forcefield at high temperatures and pressures for methane up to 298 K and 1,000 MPa in Ref. 45, which show how well TraPPE forcefield reproduces fluid volumetric properties. We also added a comment about this matter (lines 155-157). We also found in this work that TraPPE forcefield reproduces well volumetric properties of ethane, with small deviations at high pressures (Fig. 7).

Comment 2: In the methodology, the representative models should be presented graphically rather than relying solely on textual descriptions.

Response 2: We created a graphical representation of the main interaction (Fig. 1), which is the Lennard-Jones potential, used to calculate inter-chain interactions, but also 1-5 intra-chain interactions. We also added a comment to the content of this figure (lines 136-137).

Comment 3: The manuscript does not provide structural details of PE-ethane interactions. These should be included, particularly for statements such as "ultrathin metastable films at the mechanical stability limit are composed of two overlapping interface" and "interfacial tension decreases exponentially with increasing gas pressure and is primarily governed by inter-chain interactions at the interface." Local simulation snapshots or schematic illustrations are necessary to help readers construct a physical picture.

Response 3: As the reviewer suggested we included local simulation snapshots on key points. One snapshot includes the polyethylene film with adsorbed ethane molecules in the region limited in the normal direction on plots of Fig. 6. Using this snapshot the reader can have a visual image of where the limits of the interfacial regions are for one of the systems and have an idea of how many sublayers of chains in the normal direction form one interface. A description of this snapshot was also included (lines 327-330). The second snapshot shows how fractions of PE chain desorb at the interfaces inducing the accumulation of ethane molecules. This snapshot was incorporated to Fig. 10. A description of this snapshot was also included in the caption of the figure. We did not figure out how to represent graphically the following statement "interfacial tension decreases exponentially with increasing gas pressure and is primarily governed by inter-chain interactions at the interface". Therefore, we did not include any graphical content.

Comment 4: The films studied here are pristine, untreated PE. Can the obtained data be fairly compared with experimental films reported in the literature, which were pretreated with n-heptane and had unknown thickness? If not, could the conclusions be overestimating interfacial adsorption?

Response 4: The results cannot be compared fairly to films pretreated with n-heptane, the interfaces are already saturated with n-heptane. In terms of solubilities in bulk PE, references 19 and 20 (lines 53-59) show that gas pressures one order of magnitude lower are needed for but-1-ene to achieve the same solubility than ethylene, which indicates a dependence between the solubility and the molecular weight of the light gas. Therefore the n-heptane will probably saturate the interface. Even if some ethane is adsorbed at the interface the mass of n-heptane probably overweight the adsorbed mass of ethane. We do not think the results of this work are overestimated. We now included a comparison with an experimental study using CO2 and H2 gases at similar temperatures, and our results fall between those gases. This comparison is included in the text (lines 451-453).

Comment 5: The results show that the density of ethane in the PE center undergoes a reversal with increasing temperature, while interfacial adsorption also decreases with increasing temperature. The authors should provide a reasonable explanation: for example, is this reversal directly related to the melting point, crystallinity, or microstructural characteristics of PE films? Can local structural statistics from simulations support these interpretations?

Response 5: We also think this is a very interesting research area we can explore, but it will probably be developed in a new paper. Much work are needed to establish both temperatures (melting and solubility transitiion) for systems with adsorbed ethane. Other properties as cristallinity and microstructural characteristics can also be explored as source of explanation. We corrected Ref. 60, now it refers to a paper, which establishes that melting temperatures of ultrathin layers of polyethylene depends on the thickness of the film and are lower than bulk polyethylene (lines 381-382). The variability of the melting point in ultra-thin polyethylene films adds one more variable to the proposed research.

We rewrote some parts of the text, to make the text more clear. They are highlight in green.

Round 2

Reviewer 2 Report

Comments and Suggestions for Authors

Accept in current form.